

# The effect of obesity on the outcome of thoracic endovascular aortic repair: a systematic review and meta-analysis

Jiajun Li[1,*], Yucong Zhang[2,*], Haijun Huang[3], Yongzhi Zhou[1], Jing Wang[1] and Min Hu[1]

[1] Division of Cardiothoracic and Vascular Surgery, Tongji Hospital, Tongji Medical College, Huazhong University of Science and Technology, Wuhan, Hubei, China
[2] Institute of Gerontology, Department of Geriatrics, Tongji Hospital, Tongji Medical College, Huazhong University of Science and Technology, Wuhan, China
[3] Department of Urology, Tongji Hospital, Tongji Medical College, Huazhong University of Science and Technology, Wuhan, China
[*] These authors contributed equally to this work.

Corresponding author
Min Hu, Huminchn@tjh.tjmu.edu.cn

## ABSTRACT

**Background**. Obesity is a well-known predictor for poor postoperative outcomes of vascular surgery. However, the association between obesity and outcomes of thoracic endovascular aortic repair (TEVAR) is still unclear. This systematic review and meta-analysis was performed to assess the roles of obesity in the outcomes of TEVAR.

**Methods**. We systematically searched the Web of Science and PubMed databases to obtain articles regarding obesity and TEVAR that were published before July 2023. The odds ratio (OR) or hazard ratio (HR) was used to assess the effect of obesity on TEVAR outcomes. Body mass index (BMI) was also compared between patients experiencing adverse events after TEVAR and those not experiencing adverse events. The Newcastle–Ottawa Scale was used to evaluate the quality of the enrolled studies.

**Results**. A total of 7,849 patients from 10 studies were included. All enrolled studies were high-quality. Overall, the risk of overall mortality (OR = 1.49, 95% CI [1.02–2.17], $p = 0.04$) was increased in obese patients receiving TEVAR. However, the associations between obesity and overall complications (OR = 2.41, 95% CI [0.84–6.93], $p = 0.10$) and specific complications were all insignificant, including stroke (OR = 1.39, 95% CI [0.56–3.45], $p = 0.48$), spinal ischemia (OR = 0.97, 95% CI [0.64–1.47], $p = 0.89$), neurological complications (OR = 0.13, 95% CI [0.01–2.37], $p = 0.17$), endoleaks (OR = 1.02, 95% CI [0.46–2.29], $p = 0.96$), wound complications (OR = 0.91, 95% CI [0.28–2.96], $p = 0.88$), and renal failure (OR = 2.98, 95% CI [0.92–9.69], $p = 0.07$). In addition, the patients who suffered from postoperative overall complications ($p < 0.001$) and acute kidney injury ($p = 0.006$) were found to have a higher BMI. In conclusion, obesity is closely associated with higher risk of mortality after TEVAR. However, TEVAR may still be suitable for obese patients. Physicians should pay more attention to the perioperative management of obese patients.

## INTRODUCTION

Descending thoracic aortic (DTA) diseases mainly include dissections, aneurysms and traumatic injuries. These diseases can cause a serious risk of bleeding or vital organ malperfusion in acute conditions, leading to fatal outcomes.

Because of the progress of diagnostic imaging and the prolonged life expectancy of the population, the incidence of thoracic aortic aneurysm (TAA) and type B aortic dissection (TBAD) is increasing (*Kuzmik, Sang & Elefteriades, 2012*; *Howard et al., 2013*). It is estimated that AD annually affects 3–4 people per 100,000, most of whom are men over 60 years old with a history of abnormal blood vessels and/or hypertension (*Jubouri et al., 2022*). Aortic aneuryism (AA) which refers to pathological dilatation of the aorta, is considered to be the second most common disease of the aorta, subsequent to atherosclerosis (*Badran et al., 2023*).

Although significant progress in medical and surgical treatment modalities has been achieved for years, the general mortality associated with aortic diseases is still high. In recent decades, the surgical treatment modality for emergent thoracic aortic conditions has shifted from open surgery to thoracic endovascular aortic repair (TEVAR), a relatively minimally invasive modality (*Habib et al., 2023*). In particular, TEVAR is recommended as the first-line treatment for DTAA with no elastopathy and a maximal diameter of aneurysm ≥55 mm (*Isselbacher et al., 2022*). In addition, technological progress in TEVAR has also decreased the mortality rates in acute TBAD patients (*Eleshra et al., 2020*).

Unfortunately, postoperative complications of TEVAR, such as cerebrovascular diseases, arterial perforation or rupture, and renal insufficiency, may lead to higher early mortality, longer hospital stays, and a poor quality of life (*Liu et al., 2021*). In addition, reintervention may be required in some TEVAR procedures. However, the risk factors for poor outcome of TEVAR remain unclear and need to be identified.

Obesity is closely associated with adverse postoperative outcomes including wound, cardiopulmonary, and respiratory complications (*Choban & Flancbaum, 1997*). Physiologic and anatomic factors related to obesity may increase the risk of anaesthetic-related events and perioperative complications (*Alshaikh et al., 2018*). When undergoing emergent surgery, obesity may increase the risk of death caused by fulminant respiratory failure (*Choban & Flancbaum, 1997*). The elevated rate of obstructive sleep apnea in patients with obesity who received general anaesthesia may also lead to cardiovascular complications, such as stroke, myocardial infarction or ischemia, and mortality (*Choban & Flancbaum, 1997*). Surprisingly, in a systematic review, obese patients are found to have lower 30-day mortality in abdominal aortic aneurysm patients undergoing endovascular aortic repair (*Naiem et al., 2022*). However, the role of obesity in the prognosis of TEVAR has not been fully elucidated. Therefore, we conducted this systematic review and meta-analysis to summarize the roles of obesity in the prognosis of TEVAR.

## MATERIALS & METHODS

This study was performed according to the Preferred Reporting Items for Systemic Reviews and Meta-analysis (PRISMA) guidelines (*Liberati et al., 2009*). Before identifying studies

from databases, we registered this study in the International Prospective Register of Systematic Reviews (PROSPERO) on July 3, 2023 (ID: CRD42023438854).

## Literature search

In July 2023, we conducted a systematic literature search in the Web of Science and PubMed databases. Studies assessing the association between obesity and outcomes of TEVAR were assessed by full-text review. The following terms and relevant combinations were used: "thoracic endovascular aortic repair", TEVAR, BMI, "body mass index", obesity, and overweight.

## Selection criteria

The inclusion criteria included the following: (1) studies evaluating the associations between obesity and outcomes of TEVAR; (2) studies reporting one of the following results: (a) hazard ratios (HRs) or odds ratios (ORs) with relevant 95% confidence intervals (CIs) of obesity, or incidences of adverse events after TEVAR; (b) mean with standard deviation of BMI in patients who experienced adverse events after TEVAR or not; (c) incidence of adverse events after TEVAR in patients with different BMIs; and (3) clinical studies regarding adults and published in English.

The exclusion criteria were (1) editorials, commentaries, case reports, meeting abstracts, letters, or reviews; (2) studies only regarding abdominal EVAR; (3) studies reporting overlapping data; (4) studies regarding pregnant women; and (5) invalid data for pool analysis.

Based on the above criteria, the initial study screen was conducted according to titles and abstracts. The full-text evaluation was then conducted. Additional manual search for references from potential studies was employed. Two reviewers independently screened the studies (JL and YCZ). Disagreements were resolved by consulting a third researcher (HH).

## Data extraction and quality assessment

Data from the enrolled studies were extracted by two independent researchers (JL and YCZ). We extracted patient baseline characteristics and basic information of all included studies. Data regarding the relationship between obesity and outcomes of TEVAR were extracted: (1) HRs or ORs with corresponding 95% CIs; (2) mean with standard deviation of BMI; and (3) incidence of adverse events after TEVAR. Outcomes included mortality, overall complications and several specific complications.

Two reviewers (YZZ and JW) independently assessed the quality of enrolled studies by using the Newcastle–Ottawa Quality Assessment Scale (NOS), which has a maximum total score of 9 based on assessment of three domains: (1) selection of study groups, (2) comparability of groups, and (3) ascertainment of outcome of interest. Studies with a total score of 1 to 3, 4 to 6, 7 to 9 in the NOS scale were considered low, intermedia, and high quality, respectively (*Deeks et al., 2003*). A third reviewer (HH) was discussed to resolve discrepancies. Publication bias was assessed by funnel plots.

## Data analysis

We used RevMan 5.3 (The Nordic Cochrane Centre, Copenhagen, Denmark) to conduct the meta-analysis. ORs and 95% CIs were directly obtained from articles or calculated according to the incidence of adverse events in patients with or without obesity. We applied random-effects model for pooled analysis to achieve conservative results. Heterogeneity was tested according to the chi-squared test and relevant $I^2$ statistic. $p < 0.05$ or $I^2 > 50\%$ indicated significant heterogeneity. The $Z$ test was used to determine overall effects with $p < 0.05$ considered statistically significant. Subgroup analysis was performed based on specific adverse events.

Because of the variance of BMI level classifications in some studies, the incidence of adverse outcomes from the patients with the highest BMI level was extracted as the exposure group in each study.

## RESULTS

After excluding duplicate articles, three hundred and five articles were obtained in the initial search. Twenty-six articles were chosen for further full-text evaluation according to titles and abstracts. Finally, 10 articles with 7,849 individuals were included (*Romijn et al., 2023*; *Borghese et al., 2023*; *Niu et al., 2023*; *Zha et al., 2021*; *An et al., 2021*; *Naazie et al., 2022*; *Lu et al., 2020*; *Janczak et al., 2019*; *Zakko et al., 2014*; *Khoynezhad et al., 2007*). The study screening flow diagram is shown in Fig. 1. The basic information and patient baseline characteristics are summarized in Table S1, Fig. S2. Among these studies, three reported the BMI of patients who experienced adverse events after TEVAR or not (*Niu et al., 2023*; *An et al., 2021*; *Zakko et al., 2014*), five reported the ORs or HRs of obesity and the incidence of adverse events (*Romijn et al., 2023*; *Borghese et al., 2023*; *Naazie et al., 2022*; *Lu et al., 2020*; *Khoynezhad et al., 2007*), and four reported the incidence of adverse events in patients with or without obesity (*Borghese et al., 2023*; *Zha et al., 2021*; *Janczak et al., 2019*; *Khoynezhad et al., 2007*). According to the NOS, all enrolled studies were considered high quality (Table S3).

### Association between obesity and incidence of overall mortality and complications for TEVAR

Three studies (*Romijn et al., 2023*; *Borghese et al., 2023*; *Naazie et al., 2022*) reported the ORs of obesity and mortality (Fig. 2). The results indicated that the risk of overall mortality was increased in obese patients (OR = 1.49, 95% CI [1.02–2.17], $p = 0.04$). Although the heterogeneity was insignificant ($I^2 = 47\%$, $p = 0.15$) among studies, only one study showed significant association between obesity with 30-day mortality.

Three studies (*Naazie et al., 2022*; *Lu et al., 2020*; *Khoynezhad et al., 2007*) reported the ORs of obesity and overall complications (Fig. 3). The results indicated that the risk of overall mortality was increased in obese patients, though without statistical significance (OR = 2.41, 95% CI [0.84–6.93], $p = 0.10$). Significant heterogeneity was found among studies ($I^2 = 88\%$, $p < 0.001$).
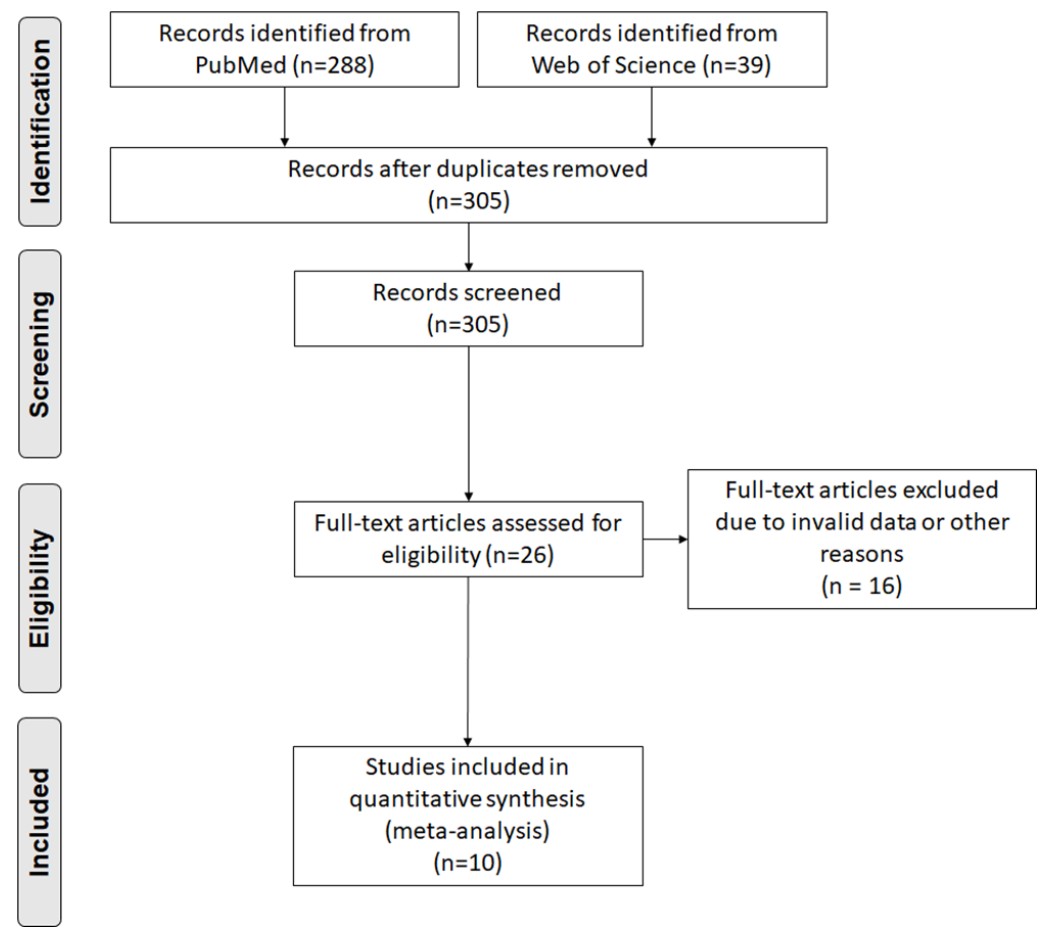

**Figure 1** Preferred reporting items for systemic reviews and meta-analysis flow diagram of literature screening.

## Association between obesity and incidence of specific complications for TEVAR

Interestingly, the associations between obesity and specific complications were all insignificant (Fig. 4 and Table S4), including stroke (OR = 1.39, 95% CI [0.56–3.45], $p = 0.48$, heterogeneity: $I^2 = 67\%$, $p = 0.05$), spinal ischemia (OR = 0.97, 95% CI [0.64–1.47], $p = 0.89$, heterogeneity: $I^2 = 0\%$, $p = 0.96$), neurological complications (OR = 0.13, 95% CI [0.01–2.37], $p = 0.17$), endoleaks (OR = 1.02, 95% CI [0.46–2.29], $p = 0.96$), wound complications (OR = 0.91, 95% CI [0.28–2.96], $p = 0.88$), renal failure (OR = 2.98, 95% CI [0.92–9.69], $p = 0.07$) and reintervention (30-day: OR = 1.80, 95% CI [0.56–5.76], $p = 0.33$; 90-day: OR = 0.77, 95% CI [0.24–2.45], $p = 0.66$; long-term: OR = 0.44, 95% CI [0.16–1.20], $p = 0.11$). Only one study reported significant association between obesity with stroke (*Khoynezhad et al., 2007*).

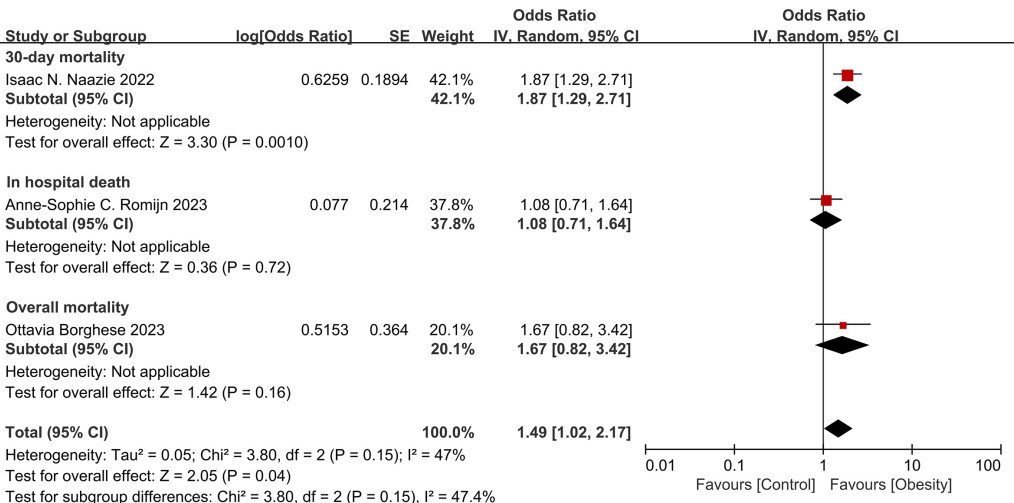

**Figure 2** Forest plot of obesity on the risk of mortality after thoracic endovascular aortic repair (*Naazie et al., 2022*; *Romijn et al., 2023*; *Borghese et al., 2023*).

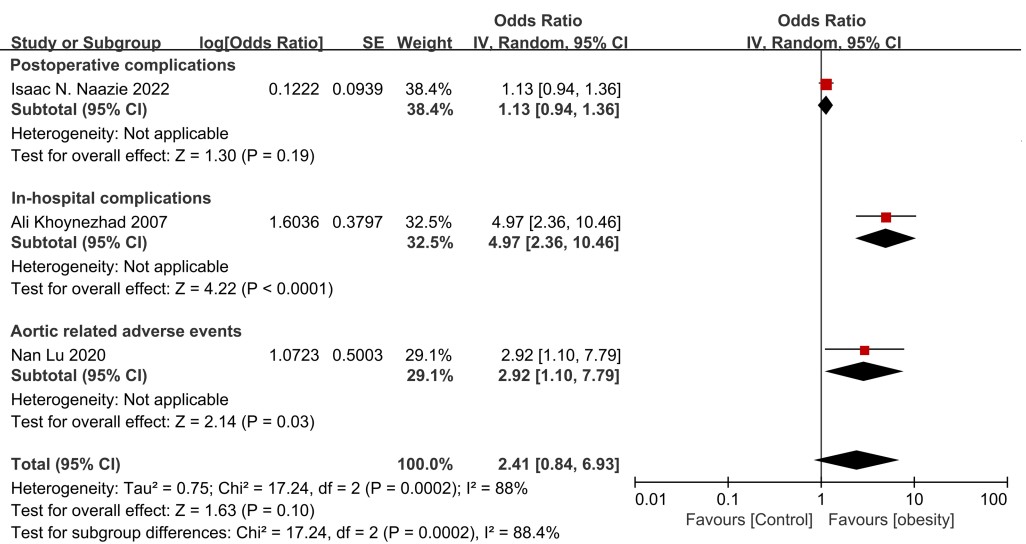

**Figure 3** Forest plot of obesity on the overall complications after thoracic endovascular aortic repair (*Naazie et al., 2022*; *Khoynezhad et al., 2007*; *Lu et al., 2020*).

## Differences in BMI between patients who experienced adverse events after TEVAR and those who did not

The patients who suffered from overall postoperative complications ($p < 0.001$) and postoperative AKI ($p = 0.006$) were found to have a higher BMI (Fig. 5). However, for patients receiving percutaneous TEVAR, BMI was not significantly associated with the success of the operation, which was defined as achieving hemostasis and maintaining limb perfusion for 30 days postoperatively without the need for exposing the common femoral artery and/or surgical repair of vessels.

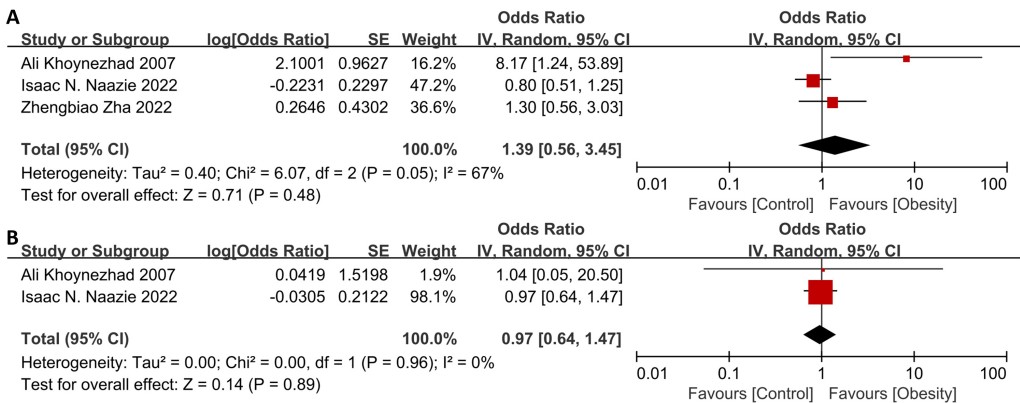

**Figure 4** Forest plot of obesity on the specific complications after thoracic endovascular aortic repair. (A) stroke, (B) spinal ischemia (*Khoynezhad et al., 2007*; *Naazie et al., 2022*; *Zha et al., 2021*).

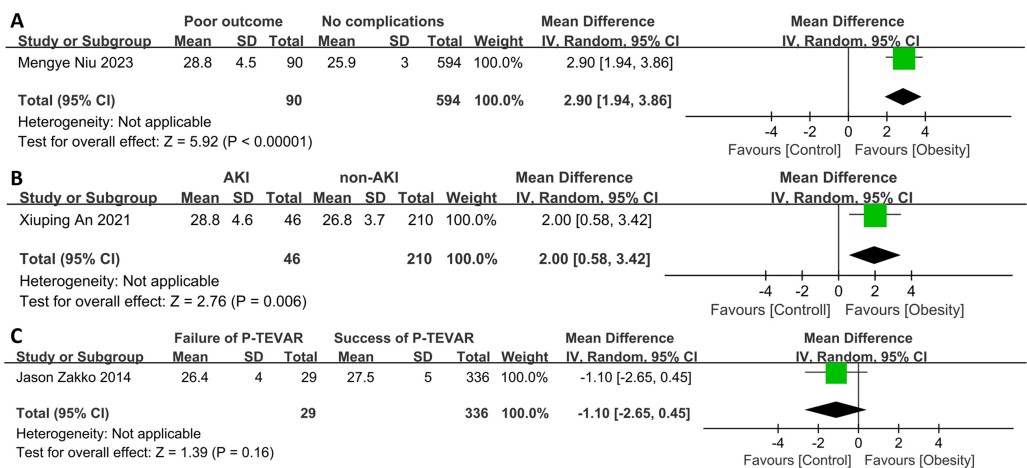

**Figure 5** Forest plot of body mass index in patients with or without. (A) Overall post-operative complications, (B) post-operative acute kidney injury, or (C) experienced successful percutaneous thoracic endovascular aortic repair or not (*Niu et al., 2023*; *An et al., 2021*; *Zakko et al., 2014*).

## Assessment of heterogeneity and publication bias

According to the funnel plots (Figs. S1–S4), significant publication bias was only found in pooled results for overall complications.

## DISCUSSION

Obesity is a well-recognized predictor for metabolic and cardiovascular diseases, which is also associated with elevated risks of postoperative general complications or even death (*Ogden et al., 2015*). First, obesity is associated with some comorbidities, including metabolic disorders (dyslipidemia, diabetes or hypertension), coronary heart diseases and respiratory disorders (sleep apnea syndrome) (*Csige et al., 2018*). In addition, obese patients also have elevated risk of pulmonary and cardiovascular adverse events as obesity

is associated with increased oxygen requirement and consumption, leading to decreased oxygenation index during general anaesthesia with impaired pulmonary compliance, and functional residual capacity (*Pelosi et al., 1998*; *Cullen & Ferguson, 2012*). In our meta-analysis, we found obesity significantly associates to high risk of postoperative overall mortality for patients receiving TEVAR.

For Stanford A dissection, obesity is also reported to be a potential risk factor for poor postoperative outcomes, which may be because of acute respiratory distress syndrome (*Aizawa et al., 2013*). Oxidative stress and chronic inflammation may be increased with obesity, as well as the levels of IL-1$\beta$, IL-6 and TNF-$\alpha$, which contribute to the development of lung injury (*Wu et al., 2020*). In our meta-analysis, obesity was significantly associated with a high risk of mortality after TEVAR, especially for 30-day mortality. It should be noted that when patients with a BMI less than 18.5 kg/m2 (underweight) were included in the control group, the associations with mortality were not significant. A meta-analysis even found that obese patients have lower 30-day mortality in endovascular aortic repair compared with nonobese patients (*Naiem et al., 2022*). The obesity paradox may provide a potential explanation. Underweight status is also reported to associate to elevated risk of morbidity and mortality for several causes, such as smoking, malnutrition, malignancy, or some wasting conditions (*Saedon et al., 2015*; *Galyfos et al., 2017*). Underweight is also associated with elevated risk of frailty or sarcopenia, which are also associated with a decline in physiologic function and a high risk of mortality in the general population and in patients receiving TEVAR (*Newton et al., 2018*; *Kim et al., 2021*; *Harris et al., 2020*). Frailty and sarcopenia, which are yet to be well-studied may also confound the result that obesity is associated with decreased risk of mortality in abdominal aortic aneurysm patients receiving EVAR (*Naiem et al., 2022*). Therefore, studies analyzing outcomes for obese, normal weight or underweight patients should be conducted in the future. More confounding factors associated with variance of weight should be adjusted. Interestingly, a study found that the operative outcomes for patients with obesity who received TEVAR vary according to presenting pathology, including DTAA and TBD (*Naazie et al., 2022*). The significant association between obesity and 30-day mortality was observed only in TBD patients. This result indicates that physicians should pay more attention to obese TBD patients receiving TEVAR in terms of perioperative management.

For postoperative complications, significant associations were not observed for overall complications nor any specific complications, including stroke, spinal ischemia, neurological complications, endoleaks, and wound complications. However, because of impaired immunity and low oxygen tension in fat tissue, obesity is also associated with a high risk of surgical site infection (*Dindo et al., 2003*; *Tanaka et al., 1993*). Nevertheless, patients who experienced overall in-hospital complications, including all-cause death and implant-related, deployment-related and systemic complications, or AKI also had higher BMI. In addition, the obesity paradox may conceal the associations between obesity and specific postoperative complications. In the study by *Naazie et al. (2022)*, underweight patients were older with elevated rates of smoking and chronic obstructive pulmonary disease, and they also have larger maximum aortic diameters, which are also reported to

be associated with adverse events after TEVAR (*Gallo et al., 2023*; *Zhao et al., 2023*; *Tang et al., 2023*; *Romeiro et al., 2021*).

The characteristics of thoracic aortic diseases and the detailed process of surgery may play more important roles in the occurrence of these specific complications than obesity. For example, the proximal landing zone location and left subclavian artery revascularization are reported to associate to elevated risk of stroke, spinal ischemia, endoleak and mortality (*Ma et al., 2023*; *Zhu, Li & Lu, 2023*; *Karaolanis et al., 2022*; *Chen et al., 2019*; *Hajibandeh et al., 2016*). Besides, the utilization of distinct anesthesia techniques can result in varied clinical outcomes among patients undergoing TEVAR. Specifically, the application of locoregional anesthesia has been significantly associated with a reduced postoperative hospital length of stay, without exhibiting any noteworthy increase in additional postoperative morbidities, compared to the administration of general anesthesia (*Panossian et al., 2023*). The clinical strategy to avoid worsening AKI in the perioperative obesity patient population is avoidance of hypotension (*Citerio et al., 2014*). However, to ensure the successful conduct of TEVAR surgery, antihypertensive medications are administered during the procedure, in a hypotensive environment, renal perfusion may be insufficient, ultimately leading to the occurrence of AKI (*Suneja & Kumar, 2014*). Therefore, the optimization of intraoperative management is paramount in the obese patient population. The precise evaluation of intravascular volume status and adequacy of fluid resuscitation poses significant challenges in obese patients undergoing TEVAR. The limited information with physical examination, and considerable variations in the accuracy of noninvasive blood pressure monitoring contribute to the complexity of these assessments. Consequently, obese patients should utilize, to the extent possible, dynamic indices of intravascular volume responsiveness to guide fluid therapy (*Balderi et al., 2008*). Additionally, the employment of goal-directed hemodynamic optimization during surgery is essential to mitigate postoperative complications (*Prowle, Kirwan & Bellomo, 2014*).

In our meta-analysis, studies that investigated the associations with overall complications or specific complications did not exclude underweight patients in the control group. Therefore, the aforementioned confounding factors should be adjusted in subsequent studies to further clearly demonstrate the role of obesity in TEVAR.

## Limitations

Several limitations of our study must be noted. First, the number of included studies is limited and publication bias was not performed. Second, because of the variance of selection criteria and patient baseline characteristics, including age and sex, heterogeneity was significant in our study. Third, the BMI range for the control group varied among the included studies. Fourth, our meta-analysis only included severe complications when analyzing the association between obesity and specific complications. Some mild or moderate complications, such as fever or infection, were not included. Fifth, the threshold for the definition of obesity varied among studies. A unified classification threshold for BMI with a high risk of postoperative complications needs to be established. Moreover, underweight patients should be excluded from the control group in subsequent studies to eliminate the effect of the obesity paradox. Sixth, due to the lack of results from multivariate

analysis, the pooled results in our study didn't adjust for covariates, which may bring bias. Last, some analysis may be dominated by one study (*Naazie et al., 2022*), which was conducted based on a prospectively collected multicenter Vascular Quality Initiative (VQI) registry data and enrolled 3,419 patients. The results of our meta-analysis should also be validated by more prospective studies. This study provides robust and convincible results by excluding underweight patients when comparing obese patients with non-obese patients. Some modalities were employed to decrease limitations. First, a comprehensive and systematic search was conducted in two databases. Second, inclusion criteria were strictly designed and utilized, which may eliminate the bias from potential confounding factors, and two independent researchers extracted the data.

### Future directions
Future studies should further investigate the associations between obesity and specific outcomes after TEVAR, such as mortality, pulmonary, cardiovascular or neurological complications. According to the results of our study, it is important to establish optimal perioperative management strategy before deciding to perform TEVAR on obese patients by comprehensively assessing the risk factors for such patients. In addition, the role of underweight in the outcomes after TEVAR should also be carefully investigated in future studies. When compare obese patients with non-obese patients, underweight patients should be excluded to avoid potential bias. Above all, there is a need for additional studies, preferably well-designed randomized controlled trials and prospective cohort studies, to compare TEVAR with other treatment modalities in obese patients.

## CONCLUSIONS
Obesity is closely associated with higher risk of mortality after TEVAR. However, TEVAR may still be suitable for obese patients. Physicians should pay more attention to the perioperative management of obese patients.

## ACKNOWLEDGEMENTS
We thank the authors of the included studies.

### Funding
The authors received no funding for this work.

### Competing Interests
The authors declare there are no competing interests.

### Author Contributions
- Jiajun Li performed the experiments, analyzed the data, prepared figures and/or tables, authored or reviewed drafts of the article, and approved the final draft.

- Yucong Zhang performed the experiments, analyzed the data, prepared figures and/or tables, authored or reviewed drafts of the article, and approved the final draft.
- Haijun Huang analyzed the data, prepared figures and/or tables, authored or reviewed drafts of the article, and approved the final draft.
- Yongzhi Zhou analyzed the data, prepared figures and/or tables, authored or reviewed drafts of the article, and approved the final draft.
- Jing Wang conceived and designed the experiments, authored or reviewed drafts of the article, and approved the final draft.
- Min Hu conceived and designed the experiments, authored or reviewed drafts of the article, and approved the final draft.

## Data Availability

This is a systematic review/meta-analysis.

## Supplemental Information

Supplemental information for this article can be found online at http://dx.doi.org/10.7717/peerj.17246#supplemental-information.

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
