# Peer review of "The effect of obesity on the outcome of thoracic endovascular aortic repair: a systematic review and meta-analysis"

_PeerJ, doi:10.7717/peerj.17246_

## Round 0.1 · original submission · Major Revisions

Please respond to the reviewers point-by-point.

**Language Note:** The review process has identified that the English language must be improved. PeerJ can provide language editing services - please contact us at [email protected] for pricing (be sure to provide your manuscript number and title). Alternatively, you should make your own arrangements to improve the language quality and provide details in your response letter. – PeerJ Staff

Reviewer 1 ·

Basic reporting

no comment

Experimental design

no comment

Validity of the findings

no comment

Additional comments

This work investigated the associations between obesity and surgical outcomes of TEVAR. The study is interesting. The authors present proper methodology. Some minor revisions need to be made.
1. The I2 statistic and the results of heterogeneity test should be reported in the Results.
2. The date of the registration of the protocol in PROSPERO should be indicated.
3. More detail information should be given on the tool used for quality assessment.
4. According to a previous meta-analysis (doi: 10.1016/j.jvs.2021.10.053), obesity is associated with lower mortality risks, which is different with current results, please reconcile or explain.
5. The potential mechanisms or reasons for higher BMI in patients suffered from postoperative AKI should be discussed.
6. The word format should be carefully checked. For example, 2 in “I2” should be presented as superscript.
7. The issue, volume and page numbers of some references seem to be missing.

·

Basic reporting

The language needs to be polished by a fluent English speaker.

Experimental design

No comment.

Validity of the findings

No comment.

Additional comments

Generally, this paper is well-organized. Here, I have minor suggestions for the paper.
1. The effect size should be provided in the Abstract.
2. Some well-known statements are made in the Introduction, which should be simplified.
3. Can authors make some analysis adjusting for covariates?
4. Full names should be provided for the abbreviations in the supplementary tables.
5. The authors should discuss detail points of perioperative management for obese patients.
6. Although data from a prospective multicenter study (VQI) were used, the results of this meta-analysis should be validated by more prospective studies.

·

Basic reporting

no comment

Experimental design

The authors conducted a meta-analysis of the impact of obesity on the outcomes after TEVAR. The results showed that obesity is closely associated with poor outcomes and emphasized the significance of perioperative management of obese patients. However, several points must be raised.
1. It seems that the authors utilized fixed effects model rather than random-effects model for pooled analysis as they claimed in the Methods section, please check.
2. Although the authors claimed that “the number of included studies is limited and publication bias was not performed”, the funnel plots still should be provided.
3. The methods for the anesthesia may influence the outcomes of TEVAR.
4. The authors should further discuss future perspectives.

Validity of the findings

no comment

---

## Round 0.2 · accepted · Accept

The authors have addressed all the concerns. The manuscript can be accepted in its current form.

Reviewer 1 ·

Basic reporting

no comment

Experimental design

no comment

Validity of the findings

no comment

Additional comments

no comment

·

Basic reporting

The manuscript is further improved. No more comments from me.

Experimental design

No comment.

Validity of the findings

No comment.

Additional comments

No comment.

·

Basic reporting

no comment

Experimental design

no comment

Validity of the findings

no comment

Additional comments

I am satisfied with the revised manuscript. It is ready for publication without any further revisions.